# Traumatic Neuroma of the Hard Palate Mimicking a Small Salivary Gland Tumor—A Case Report

**DOI:** 10.3390/biomedicines12081688

**Published:** 2024-07-29

**Authors:** Kamil Nelke, Maciej Janeczek, Edyta Pasicka, Krzysztof Żak, Marceli Łukaszewski, Jan Nienartowicz, Grzegorz Gogolewski, Irma Maag, Piotr Kuropka, Maciej Dobrzyński

**Affiliations:** 1Maxillo-Facial Surgery Ward, EMC Hospital, Pilczycka 144, 54-144 Wrocław, Poland; dr.irmamaag@gmail.com; 2Academy of Applied Sciences, Health Department, Academy of Silesius in Wałbrzych, Zamkowa 4, 58-300 Wałbrzych, Poland; kzak@ans.edu.pl; 3Division of Animal Anatomy, Department of Biostructure and Animal Physiology, Wrocław University of Environmental and Life Sciences, Kożuchowska 1, 51-631 Wrocław, Poland; edyta.pasicka@upwr.edu.pl; 4Department of Anaesthesiology and Intensive Care, Sokołowski Hospital, Alfreda Sokołowskiego 4, 58-309 Wałbrzych, Poland; marceliluk@gmail.com; 5Private Practise of Maxillo-Facial Surgery, Romualda Mielczarskiego 1, 51-663 Wrocław, Poland; nienartowicz@gmail.com; 6Department of Emergency Medicine, Wrocław Medical University, Borowska 213, 50-556 Wrocław, Poland; grzegorz.gogolewski@umw.edu.pl; 7Division of Histology and Embryology, Department of Biostructure and Animal Physiology, Wrocław University of Environmental and Life Sciences, Cypriana K. Norwida 25, 50-375 Wrocław, Poland; piotr.kuropka@upwr.edu.pl; 8Department of Pediatric Dentistry and Preclinical Dentistry, Wrocław Medical University, Krakowska 26, 50-425 Wrocław, Poland; maciej.dobrzynski@umw.edu.pl

**Keywords:** hard palate, traumatic neuroma, minor salivary glands, case report, cyst, tumors

## Abstract

In the case of any pathologies arising in the hard palate, it is always important to exclude their possible odontogenic origins. Cone-beam computed tomography is mandatory. In cases where a possible non-teeth-related pathology is confirmed, each clinician should remember possible differential diagnostics. Many small salivary glands between the mucosa and bone are present in this palatal area. Both benign and malignant tumors arising from the small glands, and mucosa of the hard palate, might occur. The case presented here mimics a solid tumor because of the nodule consistency. Because of a healthy palatal mucosa without any erosions or irritations with firm attachment to the submucosal nodule, a possible malignant tumor of small salivary gland origins was suspected in this case. When the tumor diameter is small, an excisional biopsy is required to collect good and representative material for further histopathological evaluation. In most cases, bulky nodules present on the palate are hard on palpation, non-movable, and covered with healthy mucosa. Possible bone infiltrations with mucous membrane ulcerations could manifest a more expansive character of the lesion. In the presented case, an unusual occurrence of a traumatic neuroma without any past traumatic etiology of the palate was first differentiated from the occurrence of adenoid-cystic carcinoma (ACC), pleomorphic adenoma, other benign/malignant small gland tumors, or atypical, fibroma/schwannoma of the palate. This paper presents treatment options for this rare oral neural tumor occurrence in the palate and differential diagnosis between hard palate tumors in a 42-year-old male patient, furthermore highlighting the role of an excisional biopsy as a good source for a tissue sample.

## 1. Introduction

The hard palate is located at the roof of the mouth and is composed of the palatine process of the maxillary bone and the horizontal plate of the palatine bone. Laterally, it forms the parts of the superior alveolar arch that arise from both maxillary alveolar processes. It is covered with the mucous membrane and can be divided into the anterior parts [1,2,3]. A traumatic neuroma (TN) is a rare finding within the oral cavity. A TN is most often related to a solitary nodule. Its presence can be found near vast nerve structures, like the mental foramen, tongue, or lips. Its occurrence in the hard palate is atypical and might lead to some false diagnostics [3,4]. A traumatic neuroma is not classified as a true neoplasm, but as a reactive proliferation of neural tissue related to damage and transection of some scope of trauma of a nerve bundle (0.3%). Mostly, a TN is associated with one-sided occurrence and a painless mass; however, atypical bilateral occurrence sites in the oral cavity were also reported [5,6]. This solitary nodule structure is related to the damaged nerve repair process, tissue contraction, and scarring, along with neural proliferation during the period of the healing process. A TN’s shape, size, and local symptoms might vary. Because of its firm nodular structure, a differential diagnosis with many other salivary, odontogenic, and neurogenic tumors, musculoskeletal lesions, and true neoplasms (cancers/sarcomas) should be conducted. In the presented case, any odontogenic origin was excluded because of a detailed clinical and radiological CBCT (cone-beam computed tomography) evaluation. Few cases of oral neural tumors on the hard palate are reported. Eguchi et al. reported TN occurrence in a diffuse form in the palate as a painful swelling. The study of Thomas et al. indicated TN occurrence in the oral cavity as a result of multiple surgeries in close relation to different branches of the trigeminal nerve [7,8]. In this case, however, a painless solid nodule of submucosal origins is described.

Tumors arising from the hard palate are quite challenging, especially if they are rapidly progressing to the nasal cavity or maxillary sinus, or are causing a major bone infiltration. Because the majority of tumors can manifest themselves on the hard palate, it is quite important to be aware of those that can typically or atypically manifest there [6,7,8]. Although many classifications are known, the tumor size and spread, bone infiltrations, occurrence in the oral cavity and/or adjacent areas, and co-occurring symptoms such as bleeding, bone erosions, swelling, and mucosal ulcerations might improve in the diagnostics. According to the study of Young and Okuyemi, malignant tumors of the hard palate include the following [9]: squamous cell carcinoma (SCC/SCCs), mucoepidermoid carcinoma (MEC), adenoid cystic carcinoma (AdCC), polymorphous low-grade adenocarcinoma (PLGA), low-grade papillary adeno-carcinoma (LGPA), acinic cell carcinoma (ACC), mucosal melanoma (MM), Kaposi sarcoma (KS), extra-nodal lymphomas (lymphoma/lymphosarcoma), and non-Hodgkin lymphoma (B-cell lymphoma, MALT-NHL). In addition, each of the following diseases and conditions should be differentiated: neurofibroma, neurilemmoma, schwannoma, lipoma, pyogenic granuloma, follicular lymphoid hyperplasia, necrotizing sialometaplasia (NSM), multiple myeloma, Ewing sarcoma, Langerhans cell histiocytosis, leukemia, osteosarcoma, osteomyelitis, abscess, tuberculosis, mucocele, leiomyoma, pleomorphic adenoma, and other rare benign minor salivary gland tumors, including basal cell adenoma, myoepithelioma, and cystadenoma [6,7,8,9,10,11,12]. Another review paper by Abu Rass et al. on palatal tumors divided the tumors, with the first group comprising benign lesions such as epithelial tumors of the palates—papilloma (squamous papilloma), verruca vulgaris, condyloma acuminatum, and benign pleomorphic adenoma. The second group was malignant lesions such as -MEC, ACC, PLGA, Ductal cystadenoma, acinic cell adenocarcinoma, Sialadenoma papilliferum, and adenocarcinoma NOS. The third group consisted of benign mesenchymal tumors (e.g., irritation fibroma, epulis fissuratum (inflammatory fibrous hyperplasia), peripheral ossifying fibroma, leiomyoma, rhabdomyoma, peripheral giant-cell granuloma, hemangioma, lymphangioma, pyogenic granuloma, lipoma, neurofibroma, schwannoma/neurilemmoma, granular-cell tumor, and congenital epulis). The scope of possible pathologies of the hard palate should be differentiated from those with visible mucosal lesions, those growing under a healthy mucous membrane, those with erosions/lesions on the superficial layer of the oral mucosa, and those spreading from bone or surrounding structures. The following aspects are quite important for distinguishing any possible traumatic, inflammatory, odontogenic, or other factors influencing the tumor and hard palate pathology occurrence and growth [13]. Because of this very wide possible occurrence of palatal pathologies, the authors suggest performing an excisional biopsy to gather a good volume of tissues for further histopathological evaluation. From the authors’ point of view, the occurrence of similar cases always requires an improved differential diagnosis to establish the most sufficient treatment plan possible. Because of a limited number of similar cases in the literature, this case is worth presenting, thereby improving potential readers’ awareness of such oral neural tumors.

The following case report aims to present an atypical occurrence of a traumatic neuroma (TN) of the hard palate, which occurred with submucosal swelling, lacking history or evidence of trauma, and showing an absence of mucous membrane ulcerations or odontogenic relation indicating a possible malignant salivary tumor of solitary origin.

## 2. Case Description

A 42-year-old male patient was sent for consultation because of atypical asymmetrical palatal swelling. The lesion was an atypical, painless, well-circumscribed, firm, and non-movable lump measuring about 10 mm in diameter (Figure 1). The patient’s chief complaint regarded the asymmetry and palatal swelling. On palpation, the lesion was firmly attached and hard. It was covered with a healthy palatal mucosa without any irritations or ulcerations. The occurrence time of this lesion and its growth duration were not established. Any trauma, past surgery, or other clinical anamnesis was not relevant. The patient was generally healthy, without any past illnesses. Intraoral examination excluded any possible odontogenic origin of the lesion. Both maxillary sinuses and nasal cavities were without any pathology. The left maxillary alveolar process was built properly and radiological CBCT (cone-beam computed tomography) examination did not reveal any bone erosions, infiltrations, cortical bone changes, or any other bone-related features (Figure 2). The CBCT evaluation also excluded any potential odontogenic and non-odontogenic changes within the adjacency of the hard palate lesion. Intraorally, the mucous membrane covering the lesion was firm and healthy, without any erosions or irritations, and the same color and consistency as other healthy mucosa of the oral cavity.

Because of the atypical manifestation of this tumor, the decision to conduct an excisional biopsy was made. Because of the small dimensions and possible neoplastic characteristics of this lesion, a wide excision was performed to gather sufficient tissue for microscopic evaluation. Later, depending on the final histopathological result, either a second surgery for local radicalization or just observation of the following lesion would be scheduled.

The procedure took place under local anesthesia of two ampules (1.7 mL) of Ubistesin Forte (articaine with epinephrine, 3 M, Maplewood, MN, USA). The oral cavity was rinsed with 0.1% CHX (chlorhexidine gluconate) (Eludril, Pierre-Fabre Oral Care, France) and Alfa Implant Care Mouthwash solution (Atos MM, Warsaw, Poland). No other intraoral medicaments were used. Firstly, in the distal aspect of the lesion, a hemostatic suture was applied to close the palatal artery with 4-0 interrupted vertical mattress sutures (Dafilon, B Braun, Aesculap AG Am Aesculap-Platz, Tuttlingen, Germany). The suture was placed distally from the tumor near the junction of the soft and hard palate. This suture was necessary to reduce any potential bleeding after the biopsy. The second step of the procedure was a wide excision of the tumor. The initial incision was made with a No 15c B blade (Swann Morton, WR Swann, OwlertonGrn, Hillsborough, Sheffield, UK. During the excision, both the healthy mucosal membrane and the firmly attached solid lesion were cut off towards the bone area of the hard palate. Surgical elevators were used to lift the lesion (Obwegeser 38-630-06-07- 38-630-11-07 17.5 cm/6 7/8″, KLS Martin, Tuttlingen, Germany). Because the consistency of the excised tumor was atypical, the procedure was slightly radicalized laterally to increase the surgical resection margins. The visible cortical palatal bone was smooth, without any erosion, and its color and shape were anatomical. The decision was made intraoperatively to first examine the excised tumor and then decide about the scope of future surgical intervention. The patient fully agreed to the following. Then, some horizontal mattress sutures were used to tighten the edges of the wound and ensure the stable position of the used hemostatic agent, BloodSTOP dressing (Life Science Plus, Mountain View, San Jose, CA, USA), which was used on the bone surface and the edges of the excised defect. Vertical mattress sutures were used to ensure the proper position of the hemostatic agent and reduce the bleeding (Figure 3). After the surgery, the patient was instructed to visit the clinic on the following day, and at 5 and 7 days after. NSAIDs like DexakSl 0.025 mg were prescribed for 5 days (dexketoprofenum, Berlin ChemieMenarini, Berlin, Germany), followed by the necessity of maintaining proper oral hygiene with 0.1% CHX (chlorhexidine gluconate) (Eludril, Pierre-Fabre Oral care, France) and Alfa Implant Care Mouthwash solution (Atos MM, Warsaw, Poland), which were used until the 2nd week post-operation. The post-surgical period was uneventful and good healing by granulation was achieved (Figure 4, Figure 5 and Figure 6).

Because the case was atypical, non-odontogenic, and non-trauma-related, any possible secondary surgical approach was greatly dependent on the final result from the histopathological specimen evaluation. After 3 weeks, the final histopathological evaluation revealed the traumatic neuroma of the hard palate. The histopathological result was verified with a second histopathologist. The histopathological specimen examination did not reveal a small salivary gland tumor, the lesion healed on its own fully, and no further oncologic circumstances were noted; thus, the decision was made to further evaluate and watch the lesion over time. Outcomes after 3 months from surgery were satisfactory, without any tumor reoccurrence or the necessity to perform a more radical secondary surgery (Figure 7). Further patient prognosis is, overall, very good; however, the final microscopic result was quite different from what was suspected.

Because of the atypical manifestation of an oral neural tumor in the oral cavity, the histopathological specimen was scheduled for extensive histopathological evaluation (Figure 8, Figure 9 and Figure 10). After collection, the material was fixed in a 4% formalin solution, pH 7.2–7.4, for 48 h, and then rinsed in running water for 24 h. Then, the material was dehydrated in an alcohol series and embedded in paraffin, and stained with hematoxylin and eosin. The Azan-novum method was used to demonstrate collagen fibers, and toluidine blue was used to determine the localization and abundance of mast cells. Observations were made in a Nikon80iEclipse research microscope (Nikon, Tokyo, Japan).

### Histological Findings

The results of the study highlighted that hematoxylin and eosin staining showed the presence of numerous, interwoven, overdeveloped nerves, present both in the mucosa and submucosa of the palate, surrounded by numerous blood vessels (Figure 8, Figure 9 and Figure 10). The nerves were not accompanied by lymphocytic or macrophage infiltration. It is quite interesting to also note the presence of the mast cells related to nerve damage (Figure 11).

**Figure 8 biomedicines-12-01688-f008:**
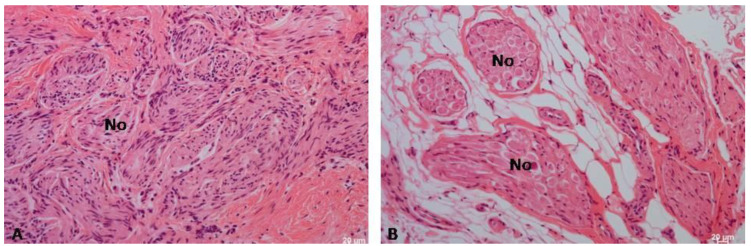
(**A**) Numerous neurons (No) visible in the mucosa. H&E, Mag. 100×, (**B**) Visible numerous neurons (No) in the submucosa with accompanying blood vessels and surrounded by fat cells. H&E, Mag. 100×.

**Figure 9 biomedicines-12-01688-f009:**
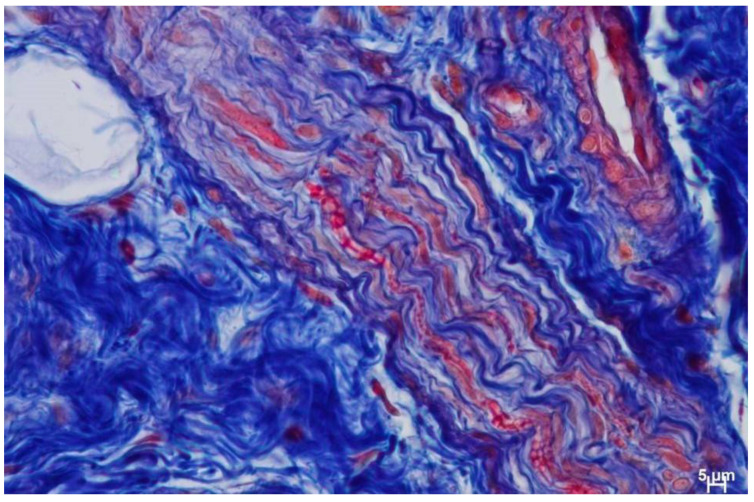
Centrally visible nerve fiber with collagen fibers present in it (blue color) and numerous blood vessels. Azan-novum, Mag. 630×.

**Figure 10 biomedicines-12-01688-f010:**
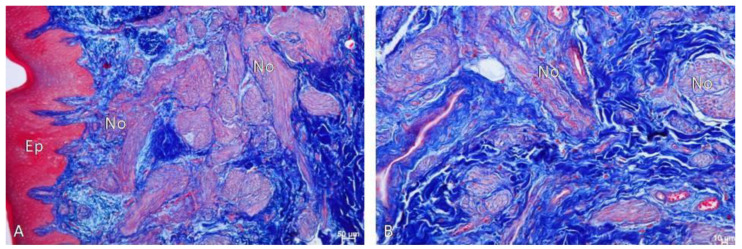
(**A**) Visible numerous neurons (No) in the mucosa covered by normal palatal epithelium (Ep). Azan-novum, Mag. 100×, (**B**) Visible numerous neurons (No) in the mucous membrane with accompanying blood vessels. Azan-novum, Mag. 100×.

**Figure 11 biomedicines-12-01688-f011:**
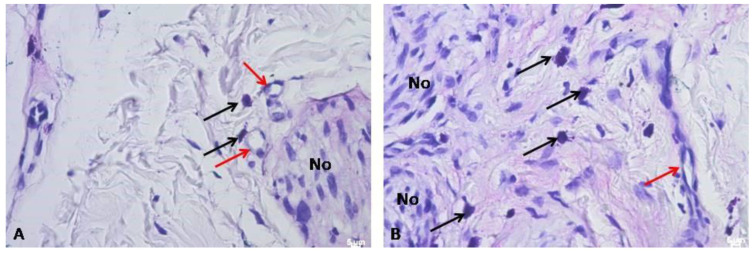
(**A**) Mast cells near blood vessels and nerve fibers above. Toluidine blue, Mag. 400×. (**B**) Numerous mast cells in the submucosa of the area. Mag. 400×. Numerous mast cells (black arrow) were found in the connective tissue of the mucosa and submucosa, accompanying blood vessels (red arrow) and nerve fibers (No). Toluidine blue, Mag. 400×.

## 3. Discussion

Some of the most aggressive tumors in the oral cavity are related to small salivary and mucous glands [13,14,15]. In the presented case, a similar lesion was suspected. Furthermore, some other possible tumors, lesions, and other medical conditions can manifest in the oral cavity and the mucous membranes [16]. Because of the great scope of each pathology, a detailed differential diagnosis is quite important. At first, any possible odontogenic-related conditions should be excluded or confirmed. This can be performed because of a good and precise CBCT and clinical evaluation. This should consist of the evaluation of the teeth (mobility, vitality) and bone (cyst, tumor spread, cortical bone condition), occurrence of soft tissue edema, crepitation, or similar factors [17]. Odontogenic cysts and tumors are the most common ones in the oral cavity; therefore, a close clinical and radio-logical evaluation is very important. The authors suggest that CBCT should be always performed, especially to investigate teeth/root involvement and the bone condition within the teeth-bearing structures.

MRI can easily exclude any malignant pathologies and nerve and surrounding tissue infiltration [18]. It is worth knowing that, in some troublesome cases, an improved histopathological evaluation is the key diagnostic tool to ensure the final and most accurate diagnosis. This case also underlines that during the excisional biopsy, each surgeon can easily evaluate the structure and consistency of the lesion, which is also helpful.

Oral traumatic neuromas (OTNs) are quite rare in the oral cavity. Their etiology is mostly associated with some damaged nerve and its healing process. Because of tumor-like features, the diagnostics are quite challenging, and they can manifest and mimic some odontogenic, musculoskeletal, sinus, and other tumors and diseases. Most commonly, they are found unilaterally; however, some rare bilateral occurrences near the trigeminal nerve fibers were also noted [19]. In this case, an atypical asymmetrical palatal swelling could be misdiagnosed with any odontogenic periapical lesion or a cyst. NT, in some cases, can be related to trauma within the oral cavity, while others with rare iatrogenic results arise with some surgeries, like Caldwell-luc procedures, teeth/cyst removal, treatment of trigeminal nerve neuralgias (nerve blockages, exheiresis, and others), oro-antral fistula/communication (OAF/OAC) treatments, or other procedures [1,2,3,4,5,6,19]. In the presented case, any possible trauma or injury to the palate was neglected by the patient. The TN should be closely evaluated and differentiated with potential non-cystic lesions, and those missing their capsule, having less visible margins, having irregular shapes and sizes, spreading towards adjacent anatomical areas, or showing perineural spreading or local invasion should be associated with low-grade malignant salivary gland tumors [17,18,19,20]. In the authors’ case study, no clinical, radiological, or other symptoms indicated this type of oral neural lesion. Histopathological evaluation is mandatory and should be conducted in each similar case. Bigger, more locally advanced tumors should also be evaluated in MR studies. Some authors, like Zheng et al., also confirm that a good MR evaluation could quite easily differentiate any possible benign, malignant, or low-grade malignant salivary gland tumors [18,19]. On the other hand, CBCT is the first diagnostic tool of choice for any dental surgeon, general dentist, or maxillo-facial surgeon. According to Ural et al., the scope of the surgical approach is greatly dependent on the tumor’s shape, size, location, anatomical boundaries, and possible aggressiveness, and the post-resection possibility of reconstruction and achieving a good nasal, sinus, and oral cavity balance without any unnecessary fistulas or communication [19,20]. The authors recommend MR for bigger lesions or to estimate the boundaries of surgical resection.

The case presented herein of a traumatic palatal neuroma is quite unusual, since no trauma, injury, or wound to the palate was confirmed, and the patient denied any trauma to the palate itself. This fact was one of the most important clinical features that resulted in a suspicion of a small salivary gland tumor of the hard palate. Although clinical judgement is very important, a good histopathological specimen evaluation can influence the final result.

When a small salivary gland tumor is not confirmed, possible oral neurogenic lesions, such as traumatic neuroma (TN) or schwannoma (SC), should be differentiated [3,21]. Others might include also mucosal neuroma, neurofibroma, palisaded (encapsulated) neuroma, or neurovascular hamartoma [3,20,21]. Currently, SC is considered an atypical, benign lesion in the oral cavity (<1%), while approx. 20–40% of each SC might be found in the head and neck region [3,20,21]. An Eguchi et al. study described a painful palatal swelling that was successfully treated with surgical excision, without any symptoms of reoccurrence [3,6,7,8,22]. It is worth pointing out that since TN is a rare lesion in the oral cavity, it can also be present in other areas of the oral cavity, and as a result, it can mimic more neoplastic lesions, rather than a typical TN or its neural variants [23].

Another atypical situation in this case is related to the occurrence of mast cells near the neural tissue. It can be assumed that the traumatic etiology initiated the appearance of mast cells in the vicinity of the nerve fibers in this case [24,25]. Because this is quite unusual, it is recommended in the case of rare or atypical tumors in the oral cavity to improve the histopathological specimen evaluation. Similar findings were noted by Kotulska et al., who found that multi-level activation of mast cells releasing various mediators has a direct impact on healing, the formation of reactions, and the form of healing in the area of the nerve endings [24,25].

The presented case indicates that the lesion was first suspected to be a small salivary gland tumor (like ACC, tumor mixtus, or similar) or perhaps a schwannoma/giant-cell tumor, but this was not confirmed. This was suspected because of a firmly attached palatal swelling, without any bone erosions or mucosal ulcerations, and the patient’s anamnesis, which revealed no past trauma, injury, or surgery/injections within the relevant palatal area. The presented case also indicates how important it is to excise a good and big enough fragment sample of the tissue to perform an adequate histopathological evaluation, which greatly influences further patient outcomes, like secondary surgery, more radical surgery, or perhaps just observation of the excised part of the palate. The presented case report is quite interesting because of the significant number of possible tumors, lesions, and conditions that can manifest on the hard palate [14].

In the authors’ presented case, the following key surgical points should be addressed:A close CBCT evaluation should exclude any odontogenic or teeth-related pathologies, cysts, and tumors in the oral cavity.A differential diagnosis of nasal, sinus, and small salivary gland tumors can be improved in any MRI studies.Excisional biopsy and gathering a good amount of studied tissue is the most effective way to investigate the nature of any oral neural tumor.Each case of a hard nodule in the oral cavity found accidentally and to be symptomless, and persisting for more than 14 days, should be scheduled for a biopsy to investigate it histopathologically.Oral neural tumors tend to mimic all other cysts, tumors, and lesions in the oral cavity.In cases of atypical lesions in the oral cavity, an improved histopathological specimen evaluation should be scheduled in order to improve the final results.

## 4. Conclusions

Any potential tumors of the hard palate require histopathological evaluation. CBCT is a good diagnostic tool for excluding any potential odontogenic cysts and tumors, while MRI can distinguish the true nature of all cysts and tumors. Oral neural tumors are a rare finding in the oral cavity, and each case requires different treatment, while surgical excision remains the treatment method of choice. The evaluation of the condition of the bone, involvement of dental roots, and adjacent mucosal membrane influence possible diagnosis. If the lesion is atypical, an excisional biopsy granting a good tissue sample is mandatory.

## Figures and Tables

**Figure 1 biomedicines-12-01688-f001:**
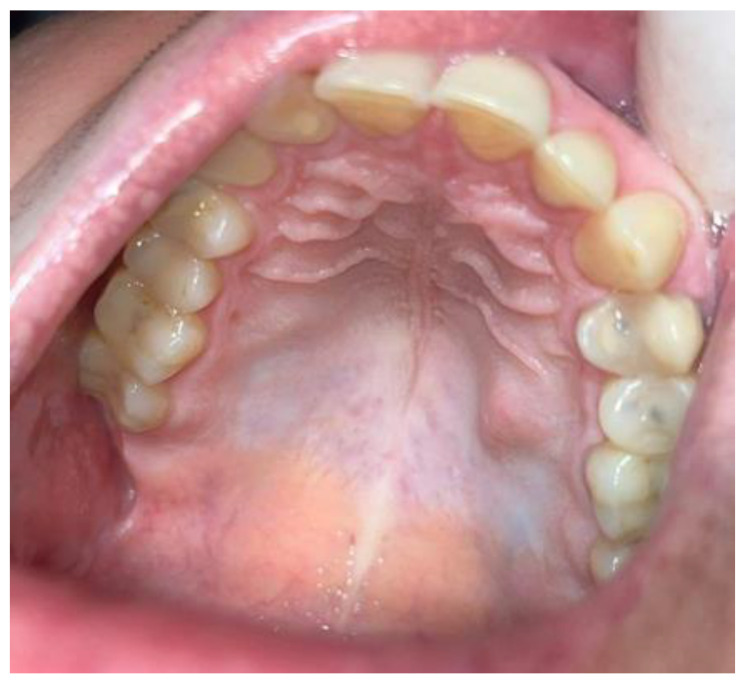
Tumor of the left side of the palate, mimicking a minor salivary gland neoplasm.

**Figure 2 biomedicines-12-01688-f002:**
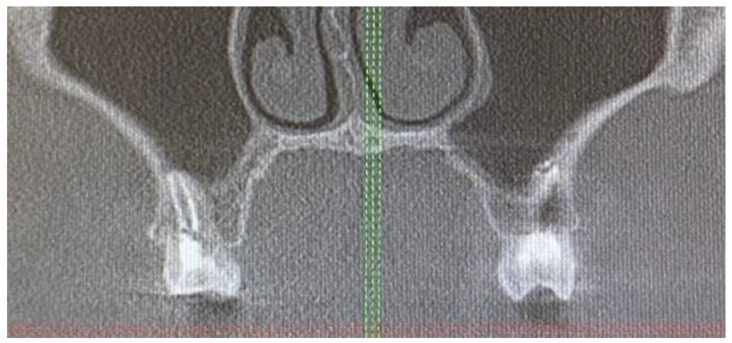
CBCT revealed no bone-related pathology. Red and green lines represent the horizontal and vertical reference lines in CBCT.

**Figure 3 biomedicines-12-01688-f003:**
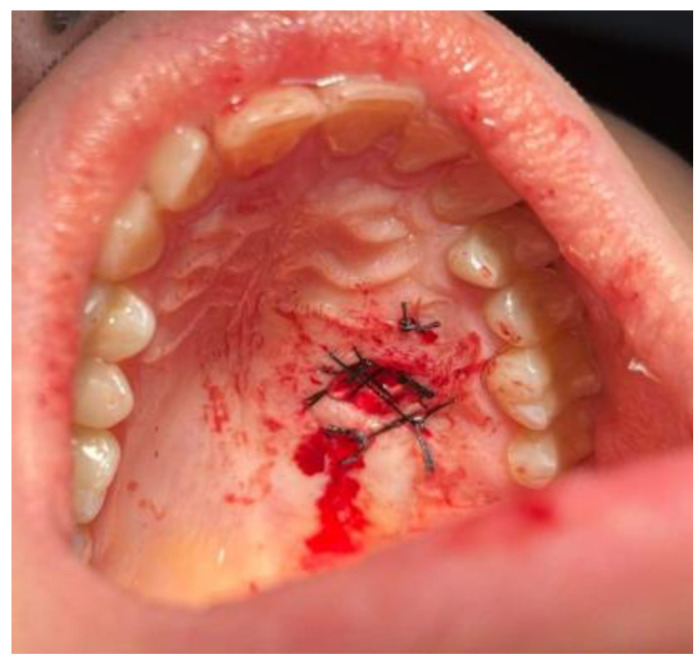
A wide excisional biopsy, followed by horizontal mattress sutures to ensure good hemostasis. Under the sutures, the hemostatic dressing is present.

**Figure 4 biomedicines-12-01688-f004:**
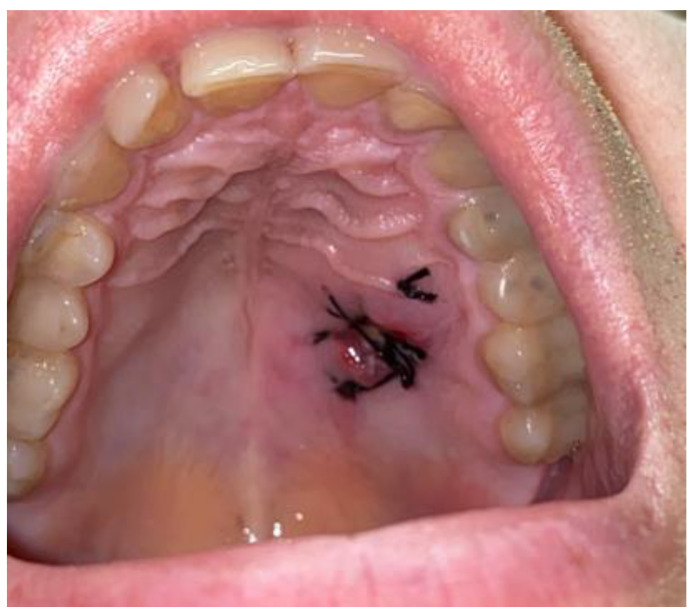
Result after 7 days.

**Figure 5 biomedicines-12-01688-f005:**
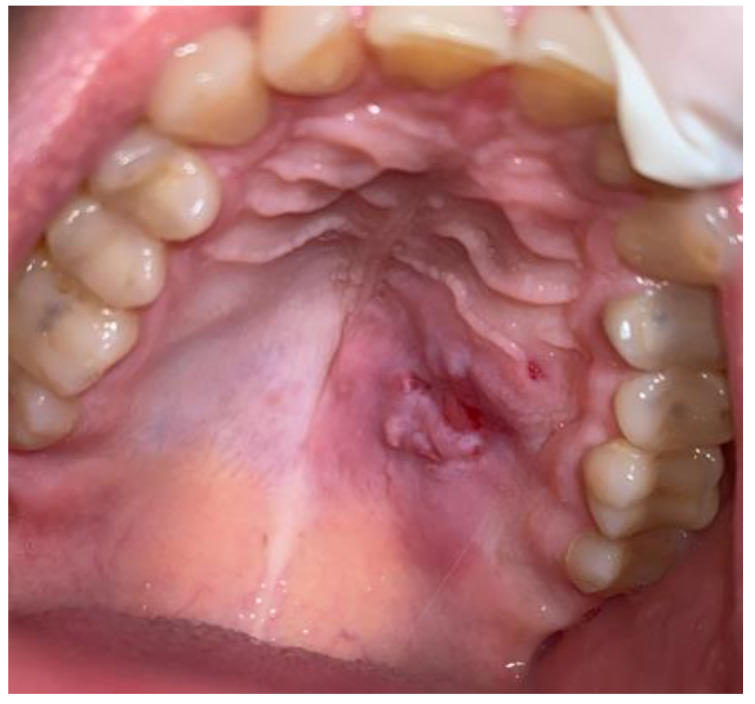
Result after 14 days.

**Figure 6 biomedicines-12-01688-f006:**
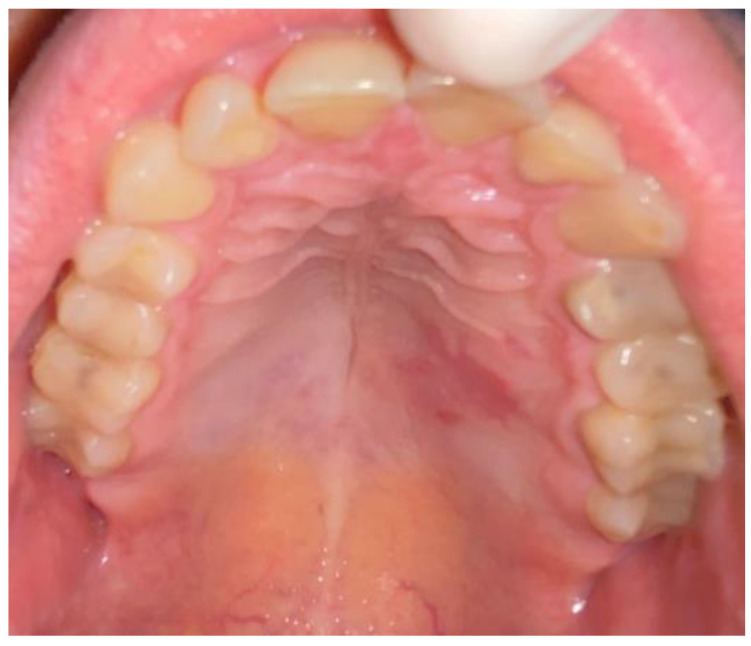
Result after a month.

**Figure 7 biomedicines-12-01688-f007:**
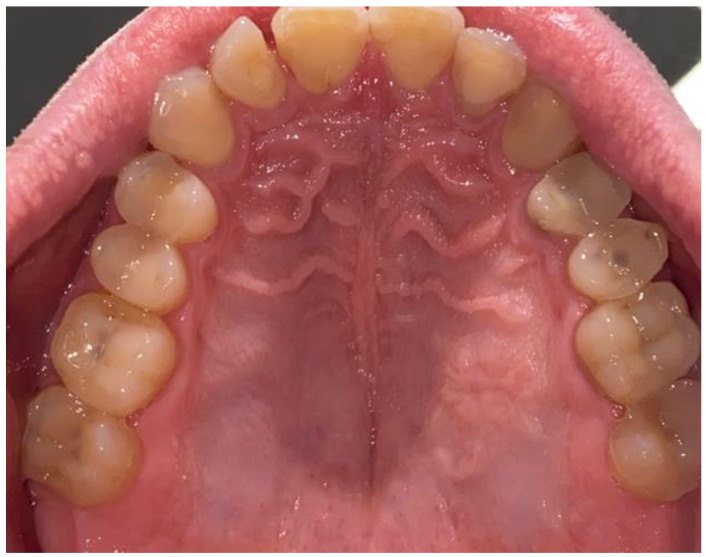
Result after three months with satisfactory outcome, without any tumor reoccurrence, progression, or secondary growth.

## Data Availability

Availability of supporting data—The datasets used and/or analyzed during the current study are available from the corresponding author on reasonable request.

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
