# Peer review of "Traumatic Neuroma of the Hard Palate Mimicking a Small Salivary Gland Tumor—A Case Report"

_biomedicines, 2024, doi:10.3390/biomedicines12081688_

Round 1
Reviewer 1 Report (Previous Reviewer 1)
Comments and Suggestions for Authors
Thank the authors for adding histology figures which are important for this case.
1. Please add a pathologist as a coauthor if it has not been done.
2. The first letters of the tumor names, like schwannoma, do not need to be capitalized.
3. Double check the spelling of words. For example, there is typo in the abstract.
4. Words should be abbreviated when they first appear in the text.
Author Response
Point 1. “ Thank the authors for adding histology figures which are important for this case.
Response 1: thank you, all advancements were made due to your valuable comments
Point 2. Please add a pathologist as a coauthor if it has not been done.
Response 2: Added – Profesor Piotr Kuropka, new co-author
Point 3. The first letters of the tumor names, like schwannoma, do not need to be capitalized.
Response 3: thank you – all changed
Point 4. Double check the spelling of words. For example, there is typo in the abstract.
Response 4: thank you – cant find much, perhaps just before the publishing the editorial team will highlight me mroe typos.
Point 5. Words should be abbreviated when they first appear in the text.
Response 5: thank you – text re-arranged
Reviewer 2 Report (Previous Reviewer 3)
Comments and Suggestions for Authors
Thank you very much for the answers.
The figures are very nice and interesting.
The authors could consider providing one more criterion to increase the quality of a clinical case, for example, following the guidelines of Gagnier et al. It is about adding to abstract, a brief description of the case (A 42-year-old male patient....) and a phrase of the conclusion.
Again , thank you.
Author Response
Point 1. Thank you very much for the answers.
Response 1: Thank you for great help in improving the paper
Point 2. The figures are very nice and interesting.
Response 2: Thank you for all your valuable comments
Point 3. The authors could consider providing one more criterion to increase the quality of a clinical case, for example, following the guidelines of Gagnier et al. It is about adding to abstract, a brief description of the case (A 42-year-old male patient....) and a phrase of the conclusion.
Response 3: thank you – dear reviewers – added : The presented paper presents treatment options for this rare oral neural tumor occurrence in the palate and differential diagnosis between hard palate tumors in a 42-year-old patient male patient, furthermore highlighting the role of an excisional biopsy as a good source for a tissue sample.
Point 4. Again , thank you.
Response 4: Also, thank you – was a pleasure working alongside.
This manuscript is a resubmission of an earlier submission. The following is a list of the peer review reports and author responses from that submission.
Round 1
Reviewer 1 Report
Comments and Suggestions for Authors
1. The focus of this manuscript is not clear. Traumatic neuroma is not a rare lesion, and the diagnosis is usually straightforward based on morphology alone. The authors described several entities for differential diagnosis, but some of them, such as adenoid cystic carcinoma, do not have morphology overlap with traumatic neuroma. If the aim of the paper is to report a common tumor in an unusual location, literature review of similar case reports will be helpful.
2. No information and figures of pathology are provided, which is essential for this case report.
Author Response
Dear reviewer thank You very much.
Response to Reviewer 1 Comments
Point 1. - The focus of this manuscript is not clear. Traumatic neuroma is not a rare lesion, and the diagnosis is usually straightforward based on morphology alone. The authors described several entities for differential diagnosis, but some of them, such as adenoid cystic carcinoma, do not have morphology overlap with traumatic neuroma. If the aim of the paper is to report a common tumor in an unusual location, literature review of similar case reports will be helpful.
Response 1: Please provide your response for Point 1. (in red) Thank you very much for the comment. This traumatic neuroma arise without any case-related or informed statement from patient of any trauma to the palate, that’s why a small salivary gland tumor was suspected. This information will be highlighted in the text. Because of many malignant salivary gland tumors arise within the palate, authors ephasize the role of differential diagnostics and wide excisional biopsy to gather sufficient amount of diagnostic tissue in order to confirm or neglect any malignant morlation, and for exaple confirm a better histopathological result which is a traumatic neuroma, without any past cases of trauma in the palate. Furthermore, aded to text:
Presented herein case of a traumatic palatal neuroma is quite unusuall, since any trauma, injury or wound to the palate was not confirmed, and the patient denies any trauma to the palate itself, This fact was one of the most important clinical feature that resulted in a suspicion of small salivary gland tumor of the hard palate.
Point 2 . - No information and figures of pathology are provided, which is essential for this case
report.
Response 2. Please provide your response for Point 2. (in red) Thank you very much. We are more than aware that no histopathological specimen photograph is presented, however its not possible to obtain such a microscopic specimen. This issue was enlisted as a potential study limitation. On the other hand the histopathological specimen photograph will not improve the paper value, since its main topic is focused on differential diagnostics and atypical features of a traumatic neuroma without any palatal trauma.
Reviewer 2 Report
Comments and Suggestions for Authors
1. The final diagnosis (oral traumatic neuroma) has not been supported. Histopathological data are neither presented (also with immunohistochemical analyses) nor discussed. A histological image would have been useful.
2. In the Discussion, the authors correctly stated, “The following cases point out that the following lesion was first suspected as a small salivary gland tumor (like ACC, tumor mixtus, or similar) or perhaps a Schwannoma/Giant-cell tumor, but none of the following was confirmed.” No data supporting this useful statement has been provided.
3. In the abstract, the authors mentioned only adenoid cystic carcinoma and fibroma/schwannoma of the palate. Why did they mention only these entities? Were they the most frequent ones on this site? Or do these entities simulate traumatic neuroma (true for schwannoma but more difficult for adenoid cystic ca)?
4. Page 2. “The third group suggested by Abu Rass et al were as “; a previous third group of “Benign Mesenchymal Tumors“ was already mentioned.
Comments on the Quality of English LanguageSome mistakes in spelling.
Author Response
Dear reviewer thank You very much.
Response to Reviewer 2 Comments
Point 1. I The final diagnosis (oral traumatic neuroma) has not been supported. Histopathological data are neither presented (also with immunohistochemical analyses) nor discussed. A histological image would have been useful.
Response 1: Please provide your response for Point 1. (in red) – Thank you, and - We are more than aware that no histopathological specimen photograph is presented, however its not possible to obtain such a microscopic specimen. This issue was enlisted as a potential study limitation. On the other hand the histopathological specimen photograph will not improve the paper value, since its main topic is focused on differential diagnostics and atypical features of a traumatic neuroma without any palatal trauma.Authors did change this study design to underline the atraumatic occurence of this truamatic neuroma, since patients anamnesis did not revealed any trauma, injury or other issues within the patients palate.
Point 2. In the Discussion, the authors correctly stated, “The following cases point out that the following lesion was first suspected as a small salivary gland tumor (like ACC, tumor mixtus, or similar) or perhaps a Schwannoma/Giant-cell tumor, but none of the following was confirmed.” No data supporting this useful statement has been provided.
Response 2: Please provide your response for Point 2. (in red) – Thank you, - statement added.
Point 3 - In the abstract, the authors mentioned only adenoid cystic carcinoma and fibroma/schwannoma of the palate. Why did they mention only these entities? Were they the most frequent ones on this site? Or do these entities simulate traumatic neuroma (true for schwannoma but more difficult for adenoid cystic ca)?
Response 3: Please provide your response for Point 3. (in red) – Thank you, - - statement added., introduction improved. In the presented case, an unusual occurrence of a traumatic neuroma wihout any past traumatic etiology of the palate was first differentiated from the occurrence of adenoid-cystic carcinoma (ACC), pleomorphic adenoma, other benign/malignant small gland tumor or atypical, fibro-ma/schwannoma of the palate. The presented paper presents treatment options for this rare oral neural tumor occurrence in the palate and differential diagnosis between hard palate tumors.
Point 4-“The third group suggested by Abu Rass et al were as “; a previous third group of “Benign Mesenchymal Tumors“ was already mentioned.
Response 4: Please provide your response for Point 4. (in red) – Thank you –issue improved, unnecessary info deleted, sentence re-written.
Reviewer 3 Report
Comments and Suggestions for Authors
The clinical case “Traumatic neuroma of the hard palate that mimics a small salivary gland tumor – case report” is interesting, but some considerations must be made.
Introduction:
To better structure the classification of exophytic lesions of the hard palate and bring together the differences and similarities of the Young & Okuyemi and Abu Rass classifications.
Description of the case:
Since the diagnosis is histopathological, the authors must provide an image of the macroscopic appearance next to the nerve and the histology. Some characteristics provided by the S-100 protein stain (or another special markers) and should be noted if there is, for example Wallerian degeneration.References.
The numbers must be noted in the text as well as their proper mention in the references section. See for example n. 19.
Thank you very much
Author Response
Dear reviewer thank You very much.
Response to Reviewer 3 Comments
Point 1. The clinical case “Traumatic neuroma of the hard palate that mimics a small salivary gland tumor – case report” is interesting, but some considerations must be made.
Response 1: Please provide your response for Point 1. (in red) – Thank you for kind words! All necessary changes are added into text.
Point 2. Introduction: - To better structure the classification of exophytic lesions of the hard palate and bring together the differences and similarities of the Young & Okuyemi and Abu Rass classifications.
Response 2: Please provide your response for Point 2. (in red) – Thank you, information added.
Point 3 - Description of the case: - Since the diagnosis is histopathological, the authors must provide an image of the macroscopic appearance next to the nerve and the histology. Some characteristics provided by the S-100 protein stain (or another special markers) and should be noted if there is, for example Wallerian degeneration.
Response 3: Please provide your response for Point 3. (in red) – Thank you, unfortunately we cannot support a histopathologcail microscopic photograph because if our histopathologist current personal situation, however we added in test that this atypical traumatic neuroma of the palate is a rare finding, becuase there was no evident trauma, injury or related damage to the hard palate.
Point 4- References. - The numbers must be noted in the text as well as their proper mention in the references section. See for example n. 19.
Response 4: Please provide your response for Point 4. (in red) – Thank you – improved, changed and general changes in the article made. Thank you.
Round 2
Reviewer 1 Report
Comments and Suggestions for Authors
I don't think this paper fits the scope of this journal. It is better to submit it to a journal focused on clinical practice, for example, otolaryngology.
Reviewer 3 Report
Comments and Suggestions for Authors
Presentation has been improved.
Thank you so much